# Genomic and Transcriptomic Characterization of Atypical Recurrent Flank Alopecia in the Cesky Fousek

**DOI:** 10.3390/genes13040650

**Published:** 2022-04-07

**Authors:** Silvie Neradilová, Alexandria M. Schauer, Jessica J. Hayward, Magdalena A. T. Brunner, Magdalena Bohutínská, Vidhya Jagannathan, Laurie B. Connell, Adam R. Boyko, Monika M. Welle, Barbora Černá Bolfíková

**Affiliations:** 1Department of Animal Science and Food Processing, Faculty of Tropical AgriSciences, Czech University of Life Sciences Prague, 16500 Prague, Czech Republic; neradilova@ftz.czu.cz; 2Institute of Animal Pathology, Vetsuisse Faculty, University of Bern, 3012 Bern, Switzerland; alexandria.schauer@vetsuisse.unibe.ch (A.M.S.); maggali.b@googlemail.com (M.A.T.B.); monika.welle@vetsuisse.unibe.ch (M.M.W.); 3DermFocus, University of Bern, 3012 Bern, Switzerland; 4Department of Biomedical Sciences, College of Veterinary Medicine, Cornell University, Ithaca, NY 14853, USA; jjh276@cornell.edu (J.J.H.); boyko@cornell.edu (A.R.B.); 5Department of Botany, Faculty of Science, Charles University, 12800 Prague, Czech Republic; magdalena.holcova@natur.cuni.cz; 6Institute of Genetics, Vetsuisse Faculty, University of Bern, 3012 Bern, Switzerland; vidhya.jagannathan@vetsuisse.unibe.ch; 7Molecular & Biomedical Sciences and School of Marine Sciences, University of Maine, Orono, ME 04469, USA; laurie.b.connell@maine.edu

**Keywords:** dog, GWAS, canine alopecia, atypical recurrent flank alopecia, Cesky Fousek, RNA-seq, differential gene expression, skin biopsies

## Abstract

Non-inflammatory alopecia is a frequent skin problem in dogs, causing damaged coat integrity and compromised appearance of affected individuals. In this study, we examined the Cesky Fousek breed, which displays atypical recurrent flank alopecia (aRFA) at a high frequency. This type of alopecia can be quite severe and is characterized by seasonal episodes of well demarcated alopecic areas without hyperpigmentation. The genetic component responsible for aRFA remains unknown. Thus, here we aimed to identify variants involved in aRFA using a combination of histological, genomic, and transcriptomic data. We showed that aRFA is histologically similar to recurrent flank alopecia, characterized by a lack of anagen hair follicles and the presence of severely shortened telogen or kenogen hair follicles. We performed a genome-wide association study (GWAS) using 216 dogs phenotyped for aRFA and identified associations on chromosomes 19, 8, 30, 36, and 21, highlighting 144 candidate genes, which suggests a polygenic basis for aRFA. By comparing the skin cell transcription pattern of six aRFA and five control dogs, we identified 236 strongly differentially expressed genes (DEGs). We showed that the GWAS genes associated with aRFA are often predicted to interact with DEGs, suggesting their joint contribution to the development of the disease. Together, these genes affect four major metabolic pathways connected to aRFA: collagen formation, muscle structure/contraction, lipid metabolism, and the immune system.

## 1. Introduction

The domestic dog *(Canis lupus familiaris)* has recently become a frequently used model organism for genomic medicine research. Over the last two decades, approximately 350 diseases have been identified that have the same or very similar underlying pathways in both dogs and humans [1,2,3]. The intensive selection during breed development, combined with a limited number of founding individuals, has resulted in long genomic regions of linkage disequilibrium within breeds. This makes association studies in dogs simpler and more straightforward than in humans [1,2]. In order to fix desirable traits, closely related founding individuals were usually bred together, resulting in high levels of inbreeding. Thus, newly established breeds often go through population bottlenecks followed by strong genetic drift that, together with inbreeding, decreases genetic diversity and increases homozygosity in populations. This leads to the fixation of undesirable traits and keeps harmful recessive alleles or dominant alleles with incomplete penetrance, resulting in late-onset and complex diseases within populations [4,5]. Genome-Wide association studies (GWAS) have proven to be a very effective tool for discovering the single nucleotide polymorphisms (SNPs) associated with a disease, even with a very low number of sampled individuals [6].

One undesirable trait observed frequently in dogs and less often in other mammalian species is alopecia or hypotrichosis, which results in the absence of the hair coat or a severely reduced number of hairs on some body regions or the entire body. Successful hair growth is dependent on the finely tuned interactions of signaling molecules and transcription factors, as well as a fully functional stem cell compartment within the hair follicle (HF). This interplay orchestrates the HF morphogenesis during the embryonic stage and the postnatal hair cycle (HC), and its coordination is dependent on complex interactions between signals of the follicular and dermal microenvironment, systemic factors, and environmental factors [7,8,9,10,11]. The HC is a lifelong process during which the HF undergoes periodic stages of growth (anagen), regression (catagen), and quiescence (telogen) [12,13]. This process relies on follicular stem cells (SCs) located in their niche [14,15]. In early anagen, cell proliferation leads to the expansion of the HF downward before the cells engulf the dermal papilla to form the hair bulb. The hair bulb is composed of matrix cells, which are highly lineage-restricted proliferative precursors. These matrix progenitors subsequently become differentiated post-mitotic cells that constitute the layers of the companion layer, the inner root sheath and the hair shaft (reviewed in [14,15]). During catagen, the next phase of the HC, the lower portion of the follicle regresses. Lastly, a state of proliferative quiescence ensues while the HF is in telogen. At any point during the next HC, but mostly during anagen, the old club hair, which is firmly anchored to the outer root sheath by trichilemmal keratin during telogen, is shed in a process called exogen [16]. Disruption of the HC results in alopecia due to either HC arrest or impaired hair quality, leading to the postnatal onset of noninflammatory alopecia.

Noninflammatory alopecia is a relatively common problem in dogs. Some forms of alopecia have a clear hereditary cause [17,18,19,20,21], and affected dogs are born without hair. In other forms, a hereditary background is suspected because of a clear breed predisposition, and these cases typically exhibit later onset of the clinical phenotype. Examples of such disorders are alopecia X [22,23,24,25,26] and recurrent flank alopecia (RFA) [27,28]. The cause of these disorders remains unknown.

Typical RFA is characterized by recurrent episodes of well demarcated, hyperpigmented areas of alopecia that affect several canine breeds, including Boxers, Rhodesian Ridgebacks, and Airedale Terriers [27,28,29,30,31]. The hair on the affected parts will usually regrow within a few weeks with recurrent loss the following year [31,32,33]. RFA is not correlated with the sex of the individual [34] and always affects adults [27]. Previous studies have shown that the onset of RFA is influenced by the photoperiod [31,34]. Histologically, RFA is associated with severely dilated infundibula filled with abundant keratin and shortened, narrowed, and often distorted proximal HF parts, resulting in the typical “witch’s feet” appearance [27]. 

An atypical form of RFA (aRFA) is anecdotally described in Vizslas, German Wirehaired Pointers, and the Cesky Fousek (CF). In this atypical form, the alopecia may still wax and wane but is more severe and affects the flanks, the sacral area, the thighs, the base of the tail, and sometimes the ears and nose (Figure 1).

With time, the atypical form may become more generalized and persistent. In contrast to the typical form of RFA, the alopecic skin is not hyperpigmented and disease onset often occurs later in life. In hunting dogs, alopecia is not only a cosmetic problem. The dogs are used for field, forest, and water work during the cold hunting season, so a dense and intact coat is vital for performance. Currently, the only option to reduce the incidence of aRFA in the breed is to eliminate affected dogs from the breeding program. The major problem with this approach is the late onset of aRFA since carriers may already have produced offspring by the time they develop alopecia. 

Our study focuses on aRFA in the Cesky Fousek (CF), a breed frequently affected by this disease. The CF is a wire-haired, versatile hunting dog breed with a small effective breeding population (*N*_e_) of 300 individuals [35]. The high frequency of aRFA in the breed has led to a reduction in its popularity, which subsequently has a further negative impact on the breed population size. On average, seven (3–11) breeding individuals are excluded from the breeding program every year (~2.5%). Considering that the CF breeding population is small, the loss of seven individuals every year is alarming. The parents and offspring of an affected individual are automatically marked as carriers of aRFA and it is forbidden to mate two carriers together.

The aim of our study was (i) to analyze the population genotypic structure associated with aRFA in the CF breed using GWAS; (ii) to establish a histological phenotype for diagnosing this poorly understood disease; and (iii) to identify specific dysregulated genes and metabolic pathways involved in the pathomechanism of the disturbed HC in animals affected with aRFA by applying transcriptome profiling to skin biopsies.

## 2. Materials and Methods

### 2.1. Blood Sample Collection

Altogether, 216 samples (non-affected n = 116, affected n = 100) were collected (189 from the Czech Republic and 27 from Cesky Fousek North America (CFNA)); 72 males and 144 females (Appendix A). Relatives were not excluded from the dataset. The blood draws were done in cooperation with the Czech Cesky Fousek Breeding Club (KCHCF) and CFNA during 2016–2019. Blood samples were shipped to the Cornell Veterinary Biobank, where DNA extraction was performed by standard salt precipitation and the DNA was then stored at −20 °C. The level of severity was determined by a responsible member of the breeding club (KCHČF). In the past, the club has developed a protocol for the identification of all aRFA levels and this protocol was also followed in our study. The Affliction of individuals was marked during sample collection, and their affected status and severity was updated throughout the study’s duration. 

### 2.2. Biopsy Sample Collection

Six-millimeter punch biopsies were taken under local anesthesia for histological evaluation and RNA extraction (seven control dogs and seven affected dogs). From the seven control dogs, two 6 mm punch biopsies were taken from sites close to each other. From the seven dogs affected with aRFA, two neighboring biopsies were taken from a completely alopecic site and two from a distant, fully haired area (shoulder). One biopsy from each site was fixed and stored in buffered Formafix 10%^®^ (Formafix AG, Hittnau, Switzerland). The second biopsy from each site was collected in RNAlater (Invitrogen, CA, USA) and stored at −20 °C until RNA extraction was performed. Based on the histological evaluation of the samples and the diagnosis of hypothyroidism in one of the dogs, we had to exclude some samples from further analysis. A comprehensive list of animals used for this study can be found in Appendix A.

### 2.3. Genotyping

Genotyping was performed on a semi-custom 220k CanineHD array (Illumina, CA, USA), currently available as the Embark genetic test [www.embarkvet.com, accessed on 19 February 2022]. In total, 216 samples were genotyped: 47 samples in 2016 with 214,582 markers and 169 samples in 2018–2019 with 239,490 markers. The number of markers differs due to the upgrading of the genotyping array. The positions of the markers are listed in CanFam3.1.

PLINK [36,37] datafiles were generated and the data were checked for errors in sex and genotype missingness. All samples had a genotyping rate higher than 95% so no samples were excluded at this point. Only markers with minor allele frequency (MAF) higher than 0.05 were included in the analyses. Data from the two different arrays were merged and discordant SNPs between duplicate samples, in accordance with a previous study [38], were removed from the datasets. Moreover, in order to avoid the resulting bias caused by the non-balanced sex of the individuals entering the analysis (Appendix A), we also filtered out the chromosome Y retrocopies [39] and sex-associated markers (a total of 96 SNPs). After filtering, the number of SNPs used for GWAS was 140,024.

Principal component analysis (PCA) was performed prior to the GWAS analyses in order to (i) check for population structure between the Czech Republic and USA samples, (ii) look for any batch effects due to the two genotyping arrays used, and (iii) identify any individual outliers. PCA was run on unlinked SNPs only, using PLINK command--indep 50 5 2, in the program EIGENSTRAT in the EIGENSOFT v5.0.1. package [40,41]. PCA plots were visualized in R i386 3.6.1 [42].

The population stratification of the data was corrected in GEMMA by including a relatedness matrix, calculated from genotypes, as a random effect. We calculated the genomic inflation factor (λ), based on *p*-values, in the *R* package snpStats [43]. Lambda inflation factor compares the median test statistic and expected null distribution and it detects the normality of the data distribution with a value of 1.0 representing no inflation. Manhattan and quantile–quantile plots of *p*-values were constructed in R. The significance thresholds for the GWAS analyses were set on Bonferroni correction on unlinked SNPs (using --indep 100 10 10 in PLINK). LD plots were created from LD analyses run in PLINK and using Matplotlib library in the Jupyter notebook [44,45]. 

### 2.4. Case/Control GWAS

A Genome-wide association study (GWAS) was conducted using a linear-mixed model in GEMMA v0.98.1 [46]. In total, 213 individuals out of 216 were used for the case/control GWAS study—three affected individuals were excluded from this analysis due to an unusual manifestation of alopecia (alopecia on the head), resulting in a dataset of 96 affected and 117 control individuals (Appendix A).

### 2.5. Quantitative GWAS (QGWAS) and Additional GWAS Analyses

In order to discover variants with a direct association with aRFA level, we performed a QGWAS analysis with 216 individuals. The three animals with the unusual manifestation of aRFA on their heads were included in this analysis. All individuals were divided into one of six phenotypic categories. The number of individuals in each category and the code of each category are: healthy (n = 111; code “0”), head affection (n = 3; code “0.1”), level 1 aRFA (n = 6; code “0.25”), level 2 aRFA (n = 28, code “0.5”), level 3 aRFA (n = 49; code “0.75”), level 4 aRFA (n =19; code “1”). Each sample was assigned to the corresponding category according to the aRFA level: Head affection—the individual loses hair on the top of the head, ears, and sometimes the top of the nose. Level 1—the individual loses hair on the ears only (can enter the breeding program); Level 2—the individual loses hair on the body sides up to the size of approximately 10 × 10 cm; Level 3—hair loss on the body sides up to approximately 10 × 25 cm; Level 4—hair loss on the body sides up to approximately 10 × 40 cm; Level 5—hair loss on the body sides larger than 10 × 40 cm (this level was not represented in our dataset).

Moreover, we conducted four additional GWAS analyses to identify specific variants that were associated with level 2 aRFA (28 individuals), level 4 aRFA (19 individuals), age of onset before 2 years of age (26 individuals), and age of onset at 6–8 years of age (20 individuals). The last two groups were also affected by level 2 aRFA or worse. We did not consider level 1 aRFA as “affected”. The control group for all the above-mentioned groups was composed of 35 individuals aged 10+ years in which the chances of developing aRFA were very low. The settings of the allelic and genotyping frequencies were the same as in the main case/control GWAS analysis.

### 2.6. Haplotype Identification

Prior to haplotype identification, we divided the genotyping data by chromosomes using PLINK v.1.9 [47,48] and subsequently phased each chromosome of interest, based on the case/control GWAS results, in SHAPEIT.v2.r837 [49]. The settings were left at their default levels with 7 MCMC burn-in iterations, 8 pruning iterations, and 20 main iterations. The number of conditioning states (K) was left at 100, the --window size setting was 2 Mb, and the genetic map was not provided, leaving the --rho value at its default (0.0004). Each phased chromosome was then transferred to PED/MAP format and run in PLINK v1.07 [37] in order to estimate the haplotypes in both the case and control groups of individuals. The setting was set to --hap-window from 1 to 10 SNPs to obtain all one-, two-, and up to ten-SNP windows across the dataset (respecting the chromosome boundaries).

### 2.7. Histopathological Analysis

Formalin-fixed biopsies were processed for routine histological analysis by embedding in paraffin, microtoming (3 μm), and staining with hematoxylin and eosin (H&E). The samples were blinded and a histopathological analysis was conducted to characterize specific histological features and patterns associated with this alopecic disease. Histological evaluation was also utilized to include or exclude samples not suitable for RNA extraction based on histological findings that might influence gene expression (e.g., secondary lesions such as inflammation). Based on this, we excluded the biopsies from one control dog, two lesional sites of alopecic dogs, and three normal skin sites from affected dogs from the analysis, resulting in a total of six samples from control animals (B2, B5, B6, B12, B13 and B14), five samples of alopecic skin (B3, B7, B9, B10, B11), and four samples of normal skin from affected dogs (B3, B9, B10, B11). Factors for exclusion included an endocrine imbalance in one of the alopecic dogs and pronounced inflammation in the biopsy of one control dog, three biopsies of normal skin of alopecic dogs, and one biopsy of alopecic skin of an affected dog.

### 2.8. RNA Extraction and RNA-Seq Experiments

RNA extraction and cDNA sequencing experiments were conducted according to the protocol outlined previously in [25,50]. All 11 samples were of high quality with a RIN > 9. After sequencing, the Illumina BCL output files with base calls and qualities were converted to FASTQ file formats and demultiplexed.

All reads that passed quality control were mapped to the canine reference genome (CanFam3.1) by STAR aligner version 2.5.3.a, as described in [25,50]. The alignment of RNA-seq reads from each sample was summarized by the number of splice arrangements per sample. The read abundance was calculated using the count software HTseq [51] and an NCBI annotated GTF (release 103) file.

### 2.9. Differential Expression Analysis

The R DESeq2 package [52] was used for differential expression analysis as described in [25]. For each gene, normalized read counts were fit to a generalized linear model (GLM) with the design formula where the condition was the factor of interest in two states: control and affected. Transcripts were considered to be differentially expressed with a Benjamini and Hochberg false discovery rate (FDR) of <0.01. The differentially expressed genes (DEGs) were mapped to biological networks using an open-source, open access, and manually curated pathway database called Reactome [https://reactome.org/ version 71; accessed on 12 January 2021]. Separate lists of upregulated and downregulated genes were uploaded separately into the database and were analyzed and matched with known biological processes and pathways.

### 2.10. Protein–Protein Interaction Analysis

We searched for potential functional associations among our GWAS and differentially expressed candidate genes using the STRING database [53], following the approach described in the study of Bohutínská et al. [54]. We were able to retrieve predicted protein–protein interactions for 132 out of 144 GWAS candidates and for 144 out of 236 strongly DEGs (exceeding the Log2FC value of +/−2). We used the ‘multiple proteins’ search in *Canis lupus*, with text mining, experiments, databases, co-expression, neighborhood, gene fusion, and co-occurrence as information sources. We used a minimum confidence of 0.4 and retained only 1st shell associations (proteins that are directly associated with the candidate protein: i.e., immediately neighboring network circles).

## 3. Results

### 3.1. Population Genetic Structure of aRFA Affected and Control Individuals

We first inquired whether aRFA individuals appear evenly distributed among populations. Even though a certain level of genetic differentiation between the Czech and the North American populations exists [55], the case and control samples were evenly distributed across the whole dataset (Appendix A); thus, we decided to use all samples for the GWAS analyses. We also checked for a possible batch effect since the samples were genotyped in different years and the genotyping array had been updated. No batch effect was identified, as shown in Appendix A.

### 3.2. Case/Control GWAS

To identify genetic variants associated with aRFA, we performed GWAS, using the presence/absence of aRFA as a predictor of phenotype. Our main within-breed case/control GWAS analysis revealed a significant association with aRFA on chromosome 19 (*p* = 1.08 × 10^−6^) (Table 1; Figure 2). Of the top ten SNPs, six are located on chromosome 8 and five of these are within the region 43,341,000–43,490,000 bp. Genotypes and their frequencies for the top SNPs on chromosomes 19 and 8 are shown in Table 2. The lambda value (*λ* = 1.01) shows that the stratification correction worked well. The significance threshold was based on the Bonferroni correction (alpha = 0.1; cut-off = 1.16 × 10^−6^). Only the chr19 association can be considered significant, while the other identified variants are considered suggestive.

The distribution of genotypes for the chromosome 19 association shows that 59% of controls are of genotype AA while only 27% of cases are of the same genotype, and nearly 19% of cases are GG compared to only 6% of controls. For the chromosome 8 association, the highest proportion of cases (70%) have the genotype GG compared to only 44% of controls. 

### 3.3. Quantitative GWAS and Additional GWAS Analyses

Quantitative GWAS (QGWAS) and additional GWAS analyses were performed in order to find possible variants associated with the aRFA level of severity represented in our dataset, as well as the age of aRFA onset. A QGWAS analysis of six phenotypic categories showed that seventeen of the twenty top SNPs were on chromosome 8 (Table 3; Appendix A). Seven of the top ten SNPs overlapped with those identified in the case/control GWAS. Although the significance level of the case/control GWAS was not met, we considered these suggestive associations relevant as well. The significance threshold was based on the Bonferroni cut-off (alpha = 0.05) for all analyses mentioned in this section.

The genotypes of each phenotypic group for the top SNP (chr8, BICF2P361090) are presented in Table 4. Some groups consist of low sample numbers, and thus we cannot draw any definite conclusions (“head” and “L1”). Groups “healthy” and “L2” show the ratio of individuals with AA and CA genotypes close to 50%, while groups “L3” and “L4” show that most individuals carry the genotype AA. The CC genotype exhibits a comparatively lower frequency in all groups (Table 4).

For the additional GWAS analyses, we found associations with aRFA onset before 2 years of age (Table 5, Appendix A) and level 4 aRFA (Appendix A, Appendix A). The top SNP in both analyses was on chromosome 21 (BICF2G630640798) with raw *p* = 5.01 × 10^−7^ and *p* = 1.28 × 10^−6^, respectively (Table 5, Appendix A). Moreover, in the analysis of the early onset before 2 years of age (Table 5), the result shows a stronger association (*p* = 5.01 × 10^−7^; Bonferroni cut-off = 5.8 × 10^−7^) than the most significant SNP in the case/control GWAS (*p* = 1.08 × 10^−6^; Table 1). The results of the GWAS analyses of individuals older than 6 years and level 2 aRFA showed no significant associations. The average genomic inflation factor for all four additional GWAS analyses was 1.02 (range 1.00–1.05).

### 3.4. Haplotype Identification

In order to adequately extend the area on chromosomes where the candidate genes could be located, we conducted a haplotype analysis. Based on the results of the case/control GWAS we looked closely at the haplotype distribution on chromosomes 19, 8, 30, and 36. Table 6 shows the results for the most significant haplotypes and the most significant haplotypes containing the most significant SNPs (from the case/control GWAS). On chr19, 4443 SNPs passed filtering and the most significant haplotype, consisting of the motif ATGGTCAGGG (*p* = 2.09 × 10^−11^), was found in 84% of cases and 54% of controls. A single-base (A or G; *p* = 2.03 × 10^−6^) haplotype containing the top chr19 SNP from the case/control GWAS study was found, with the A-haplotype in 55% of cases and 77% of controls, and the G-haplotype in 45% of cases and 24% of controls. On chr8, 6105 SNPs passed filtering and the most significant haplotype (AAG; *p* = 7.31 × 10^−8^) was found in 75% of cases and only 49% of controls. A haplotype containing the suggestive chr8 SNP from the case/control GWAS was found (GGG; *p* = 2.25 × 10^−7^) in 85% of cases and 62% of controls. On chr30, 3922 SNPs passed filtering and the most significant haplotype (GCGA; *p* = 5.04 × 10^−6^) was found in 16% of cases and 36% of controls. A haplotype containing the suggestive chr30 SNP from the case/control GWAS has the motif ATACAGGA (*p* = 1.45 × 10^−5^) and was found in 22% of cases and 41% of controls. On chr36, 2746 SNPs passed filtering and the most significant haplotype (CC; *p* = 3.5 × 10^−5^) was found in 37% of cases and 19% of controls. This haplotype also contains the suggestive chr36 SNP from the case/control GWAS.

### 3.5. Candidate Genes Identified by GWAS

Using the results of the different GWAS analyses with subsequently constructed LD plots (Appendix A) and the abovementioned haplotype analyses, we identified 144 potential candidate genes within a 2–4 MB window using the most significant and several suggestive SNPs on each chromosome (11 genes on chr19, 61 genes on chr8, 60 genes on chr30, and 11 genes on chr36) (Appendix A and Appendix A). Given the nature of aRFA and the available scientific information regarding RFA and other non-inflammatory alopecic disorders in dogs, we focused mainly on genes associated with circadian rhythm and keratin metabolism. We identified eight genes that met these criteria (*CSNK2A1*, *PIF1*, *RORA*, *TCF12*, *FUT8*, *ZFP36L1*, *RNF111*, *SNX22*) [www.genecards.org; www.pathcards.genecards.org, accessed on 19 February 2022]. The mRNA expression of four out of the 144 GWAS candidate genes have been previously associated with different HC stages (telogen—*GULP1*, anagen—*PCLAF*, *PIF1*, *TLN2*) [56]. A spreadsheet summarizing all GWAS candidate genes is shown in Appendix A.

### 3.6. Histopathological Phenotyping and Sample Selection

To identify a precise histological phenotype of aRFA, we examined skin biopsies of aRFA affected and control dogs. The histological phenotype in all biopsies from the control dogs as well as samples of unaffected haired skin from alopecic dogs was histopathologically unremarkable (Figure 3A). HFs were predominantly in anagen, and the inferior portion of the HFs extended deep into the panniculus. Few follicles were in telogen or kenogen. Sebaceous glands appeared normal, and the epidermis was unremarkable.

All biopsies from the affected skin of alopecic animals displayed similar features previously described in typical RFA cases (Figure 3B–F). Anagen follicles were absent. Infundibuli were moderately to severely dilated, sometimes appeared long, and were filled with abundant orthokeratotic keratin, which was laminar to compact and extended into the openings of the secondary follicles, resulting in a “witch’s feet”-like appearance (Figure 3E). The follicular parts proximal to the infundibula were shortened and limited to the dermis (Figure 3B–E). In some sections, rare telogen (Figure 3C) or kenogen (Figure 3F) follicles could be identified but often only the outer root sheath was visible and a definitive follicular stage could not be assigned (Figure 3B–D). Atrophic follicles were present in some biopsies (Figure 3F). A mild distortion of the HFs could be observed (Figure 3B–F). The sebaceous glands appeared multifocally prominent. The epidermis was mildly hyperplastic and covered by mild to moderate basket-weave, orthokeratotic keratin. Excessive pigmentation was not seen.

### 3.7. RNA Sequencing Analysis

Single-end sequencing of the fifteen RNA libraries produced a mean number of 37,377,469 million (M) reads per sample on average (range: 31,686,907–43,209,034 M). The mean percentage of reads uniquely mapped to the genome was 90.19%, ranging from 87.50–91.59%. Among those, 80.84% on average mapped to the annotated canine transcriptome (range: 75.65–86.25%), resulting in 26 M counts per sample on average.

To identify genes that were differentially expressed in alopecic and healthy skin, we conducted a transcriptome analysis. A PCA plot was constructed based on gene expression profiles (controls n = 6; normal skin of affected dogs n = 4, alopecic skin of affected dogs n = 5) and demonstrates distinct clustering of samples from control dogs and biopsies of unaffected skin from affected dogs compared to alopecic skin of dogs with aRFA (Figure 4). Based on these clear clustering results, we combined samples from control dogs and healthy skin samples from dogs affected by aRFA (n = 10) and compared those with samples of affected skin from dogs with aRFA (n = 5) for further analysis.

We identified a total of 1435 DEGs with an adjusted *p*-value of <0.01. Of these, 669 genes (46.6%) were downregulated, whereas 766 (53.4%) genes were upregulated in alopecic skin samples from affected dogs (Appendix A). Of all deregulated genes, 135 were strongly upregulated with a log2fold change of at least 2 and 101 genes were strongly downregulated with a log2fold change of at least −2 (Appendix A). Twenty-five of the DEGs have previously been associated with HF morphogenesis or HC in the literature (Table 7). With the exception of *DLX5*, *LGR6*, and *NFATC2IP*, all of the HC-associated genes were downregulated and most of them were associated with the WNT or SHH (Sonic Hedgehog) pathway. Only four genes could be considered strongly downregulated, however, on the lower end of the scale (Log2FC ranging between −2.00 and −2.89; Table 7).

An analysis of the deregulated genes showed that some (n = 12) were associated with vitamin D and steroid hormone metabolism; however, of these only HSD3B2 could be considered strongly upregulated (Table 8).

### 3.8. Functional Classification of DEGs

The online Reactome pathway analysis revealed that amongst the genes downregulated in the skin biopsies, there is an overrepresentation of pathways involved in the organization and assembly of the extracellular compartment and signal transduction. For the latter specifically, 17 downregulated genes identified were involved in the SHH and WNT signaling pathways (Table 7). Pathway analysis of the upregulated genes identified that 251 genes (33%) were involved in metabolism generally, whereas 135 (18%) of the upregulated genes were specifically related to the metabolism of lipids. Among the most relevant pathways identified for up and downregulated genes, no common pathways were found (Appendix A, respectively).

### 3.9. Protein–protein Interaction Analysis 

To discover functional connections between GWAS candidate genes and RNA-seq deregulated genes, we conducted a protein–protein interaction analysis in STRING. The GWAS candidate genes showed enriched protein–protein interactions. Specifically, they were predicted to be connected in six clusters of more than three proteins each (Appendix A), altogether comprising 96 interaction edges (significant enrichment of interactions; observed N edges = 96, expected N edges = 20, *p*-value < 1.0 × 10^−16^). We identified very limited overlap between the 144 GWAS candidate genes and the 1435 DEGs in the skin biopsies. Only 11 identical genes (*SLC25A3*, *GTDC1*, *ARG2*, *PAPLN*, *RAD51B*, *RDH11*, *HACD3*, *LACTB*, *SNX1*, *TPM1*, *TRIP4*) were identified in the GWAS study and RNA-seq experiment and their overlap was not significant (*p*-value = 0.61, Fisher’s exact test; Appendix A). Moreover, no functional connections between the 11 identical genes were found. Due to the lack of overlap between GWAS candidates and DEGs, we speculated that they may be linked at the level of functional processes. This would mean that genetic changes identified in the GWAS analysis resulted in gene expression shifts of the interacting protein partners. Thus, we sought evidence that GWAS candidate genes d with the DEGs, which would identify the molecular processes related to aRFA. We took advantage of protein interaction information from the STRING database, which provides an estimate of proteins’ joint contributions to a shared function [53]. For each GWAS candidate, we searched for the presence of STRING interactors with proteins coded by DEGs. To focus only on genes with a likely stronger impact on aRFA, we limited our list of DEGs to those strongly deregulated in the skin biopsies, exceeding the Log2FC value of +/−2. Following this approach, we found that out of the 144 GWAS candidates, 40 were predicted to interact with at least one of the 236 strongly DEGs. In fact, thirteen interacted with more than one strongly DEG (Figure 5).

Several large STRING clusters were identified in this analysis (Figure 5 and Appendix A). The largest of these clusters were centered on collagen formation, muscle structure/contraction, the immune system, and lipid metabolism. Some of these cohorts were already seen among the most significant functional pathways of deregulated genes (Table 7 and Table 8). Taken together, STRING analysis supports our hypothesis of distinct molecular functions being involved in the pathogenesis of aRFA.

## 4. Discussion

Atypical recurrent flank alopecia in the Cesky Fousek is a disease that is influenced by genetic factors. Pedigree analysis shows that aRFA is more prevalent in some families than in others. Dostál et al. [57] suggest that aRFA in the Cesky Fousek is a recessive disease with incomplete penetrance. They conducted a simple statistical analysis of the offspring of parents that were affected, healthy, or a combination of both [57]. It is believed that the incomplete penetrance is dependent on environmental factors, such as housing and nutrition. 

In this study, we identified two significant associations with aRFA in the Cesky Fousek (chr19 and chr21) and other suggestive associations on chromosomes 8, 36, and 30. The suggestive associations were used for the candidate genes’ identification because these variants may sometimes help us to get more complete information about the connection of the genes to the phenotype [58]. The significance of the GWAS-identified variants in our study is comparable to other GWAS studies of complex genetic diseases in dogs (e.g., lymphoma, elbow dysplasia, mast cell tumor) [2], suggesting that aRFA likely has a polygenic inheritance. Even though we used a within-breed design, our dataset was rather small for a complex disease GWAS. Follow-up analysis using a larger sample size is needed to confirm these findings. Interestingly, the region identified in our study on chr19 maps to chr2 (144,837,140–147,020,527bp) in the human genome (hg38), and this region has been associated with male pattern baldness [59]. However, the significance of this association was relatively low (*p* = 5.65 × 10^−10^; 181 out of 287 associated regions), and thus there is a possibility that the overlap might have happened by chance.

In total, we identified 144 GWAS candidate genes based on significant and suggestive associations on chromosomes 19, 21, 8, 30, and 36. Genotype analysis for the chr19 variant (BICF2G630255452) did not reveal a clear pattern between cases and controls; nevertheless, most of the control individuals were of genotype AA while most of the cases were of genotype GG and GA, suggesting the G allele is associated with higher risk for aRFA. The frequency of genotypes on chr8 (BICF2P361090) identified by QGWAS suggest that the CC genotype is associated with a lower risk of aRFA occurrence. Proportionally, more individuals severely affected by disease (level 3 aRFA and level 4 aRFA) were of genotype AA, while mildly affected (level 2 aRFA) and healthy individuals were of genotype CA. The genotype could be a contributing factor and, along with environmental factors, may influence the severity of the disease.

There are several limitations to the current GWAS study. One of them is the uncertainty of the development of aRFA in individuals from the control group. aRFA may manifest later in life; therefore, some individuals may later be reclassified. Ideally, we would use only animals aged 10+ years; however, due to the small population size of the breed this was not possible. We believe that the lack of an age threshold does not strongly affect the results of our study, because the average age of aRFA onset for our affected group was 3.9 years and out of 117 controls only five were slightly under this average age of manifestation. Another possible limitation is that the causative variants might be fixed in the population or at a high frequency; thus, the GWAS method would not detect them in a similar manner to the detection of obsessive compulsive disorder in Doberman Pinchers [60]. In this case, the identified significant and suggestive associations may only be modifiers of the causative variants. We also need to consider that the Bonferroni cut-off for the case/control GWAS is rather high. There is a discussion about multiple testing corrections and which cut-off threshold to use in order to find truly significant results [61]. Although there are several concerns about GWAS in general, it is still considered the best method for detecting associations between SNPs and hereditary diseases, which can lead to the identification of possible causative genes and variants [61]. In order to overcome the limitations of the current study, it is necessary to conduct future research with more individuals, ideally from multiple dog breeds.

Transcriptome analysis revealed 1435 deregulated genes and the vast majority of these genes are also present within microdissected anagen and/or telogen HFs [56]. Only 43 (3%) of the deregulated genes in the alopecic skin biopsies of dogs with aRFA have not been identified in microdissected HF, suggesting that these genes are derived from the HF macroenvironment. The HF macroenvironment is gaining more and more attention and it is well known that the cyclical regeneration of the HF is not only controlled by factors derived from the follicular microenvironment but also from the dermal macroenvironment [7,62,63,64].

Among the deregulated genes in our study, 25 (1.7%) genes can be clearly identified as being involved in HF development, specific HC stages, follicular SCs, or the HC. Most of them are related to the WNT and SHH pathways, which are known to be important for anagen induction, promotion, and differentiation [7,14,65,66,67,68]. Recently, Alopecia X, another noninflammatory alopecic disorder with a presumed hereditary background, was also connected to altered WNT and SHH pathways [25]. Specifically, in our study, thirteen genes (*CTNNB1*, *CUX1*, *DLX2*, *DLX3*, *CTNNB1*, *HOXC13*, *FOXN1*, *FZD2*, *FZD3*, *LEF1*, *LHX2*, *LGR4*, and *LGR5*) encoding transcription factors or signaling molecules of the WNT signaling pathway were all downregulated in affected skin samples. Equally, genes associated with the SHH pathway, including *FOXE1*, *GLI1*, *HHIP*, *SHH*, and *SMO*, were also downregulated. Among the genes involved in inhibiting anagen induction and in the BMP signaling pathway [14], only *DLX5* was upregulated. Furthermore, *MMP7*, an antagonist of the WNT pathway, was also upregulated, indicating that there is likely an active inhibitory component of the WNT pathway involved in the HC arrest observed in aRFA [69]. Interestingly, genes associated with the HF stem cell niche, namely *GLI2*, *LGR5*, *LHX2*, and *FOXE1* are all downregulated suggesting that impaired SC function is associated with the development of aRFA [14,70,71]. It has been shown that HC is dependent on a fully functional SC compartment [25,72] and it is possible that an altered SC compartment might be responsible for the extreme short follicles seen in histology. In conclusion, the findings of the RNA-seq experiments are compatible with the results of the histopathological examination showing a lack of anagen HFs and shortened follicles. The exact HC stage reflected by these shortened follicles cannot be assigned morphologically. Eventually, the downregulated genes associated with follicular SCs result in a completely dysfunctional HC that does not allow clear HC stages to be defined. This would also be supported by the histological observation of numerous HFs that have a dystrophic appearance. It is, however, still unclear whether the deregulated genes are the cause or the consequence of the HC arrest.

Eleven identical genes were identified in both the GWAS and the transcriptome analysis. While these concordant findings might imply that they represent the core genes involved in aRFA, we see two reasons that this is not necessarily the case. First, these genes were not predicted to interact, suggesting that they have roles in very different molecular processes. Second, although the 11 genes were classified as differentially expressed based on the expression significance, their actual expression shift was negligible (Log2FC ranging between 0.31 and 1.04), suggesting only a subtle impact of their differential expression on the organism.

In addition to the abovementioned DEGs associated with the WNT, SHH, and BMP signaling pathways 236 DEGs identified in the skin biopsies were predicted to interact with 40 genes of the GWAS study using the STRING database (Figure 5). They were associated with collagen formation, muscle structure/contraction, Lipid metabolism, and immune metabolic pathways (Figure 5). We identified significantly more interactions than expected in the network (enrichment *p*-value < 1.0 × 10^−16^). In Figure 5, several regulators of different molecular processes are present, such as DNA repair (PCLAF aka KIAA0101), cell cycle (CCNB2), cellular and intracellular trafficking (SLC10A1, HERC1, RAB11A), cell signaling (ITGAV, RGS6, PSEN1), etc. Some of these regulators might be responsible for the altered function of the interacting DEGs.

Lipids have been implicated in three possible mechanisms for disrupting hair growth [73]. Two of the mechanisms are unlikely in aRFA but one mechanism suggested in this review, namely that an inherently altered lipid metabolism state may be linked to the HC by affecting signaling proteins involved in the SHH or WNT pathways, is a possibility [74]. This assumption is supported by the fact that several genes encoding for molecules involved in these pathways are deregulated in the skin biopsies of dogs with aRFA. If these signaling pathways are impaired, due to, for example, a sterol precursor accumulation, the induction and promotion of the anagen HC phase is impossible, resulting in alopecia [75]. It was found that obesity had a negative impact on HF SCs and can cause hair thinning [76,77]. In our study, 30 out of 100 affected individuals (30%) were identified as overweight, while only 10 out of 116 controls were overweight (11.6%; Appendix A). Obesity could be a contributing factor to aRFA in some individuals. Amongst other lipids, cholesterol is of particular importance for the skin. It is crucial for keratinocyte differentiation, has an important barrier function, and is a precursor for steroid hormone synthesis in the skin [78]. Interestingly, in the skin biopsies of dogs with aRFA seven deregulated genes are encoding proteins, mainly enzymes, involved in sex hormone or cholesterol biosynthesis.

Cholesterol (7-Dehydrocholesterol) is also a precursor of vitamin D3 (cholecalciferol) under UV radiation [79]. Vitamin D3 is important for the skin. A mutation of Vitamin D receptors has been previously connected to Alopecia Totalis and a knockout of vitamin D receptors in mice stopped the initiation of the new HC [63,78,80,81,82]. In our study, we identified several downregulated genes associated with the vitamin D metabolism in affected skin samples (Table 8). Vitamin D has been shown to play an essential role in the biosynthesis of estradiol in mice and pigs [83]. It is well known that keratinocytes are the primary source of vitamin D and its active metabolite is processed in the skin, supporting a local deregulation of the estrogen metabolism partially mediated by vitamin D [81]. Thus, it might be interesting to explore the role of cholesterol on the HC further to identify new drugs targeting the control of cholesterol in the skin. Interestingly, genes associated with sex hormone metabolism were also downregulated in the skin biopsies of dogs affected by aRFA. The degree to which sex hormone biosynthesis and metabolism, which involves the hormones and enzymes of the complex hypothalamic–pituitary–gonadal axis, is associated with vitamin D metabolism and is involved in the pathogenesis of aRFA remains to be further explored. A disrupted sex hormone metabolic pathway and deregulated vitamin D metabolism have also been identified in another alopecic disorder with a most likely hereditary cause [25]. Future studies evaluating the nutritional and hormonal status of affected vs. control dogs would be helpful to gain more insights into the role played by these pathways.

The muscle structure/contraction metabolic pathway that was identified as another relevant pathway affecting aRFA might be associated with the arrector pili muscles (APM). The APM has recently gained attention since it inserts close to the SC region of the HF, has been associated with impaired HF cycling [84] in humans, and is associated with impaired SCs function with age [85]. Conversely, the SCs of the HFs express genes facilitating the formation of tendons and ligaments and establish a niche for smooth muscle myoblasts that create the APM [86,87]. The overexpression of these genes result in a poor vascular and nerve supply of the SC niche and contributes to SCs quiescence [87]. Our results show mostly a strong upregulation of genes in the muscle structure/contraction pathway. Moreover, genes encoding follicular SCs in the skin biopsies were downregulated further, indicating that in dogs with aRFA this pathogenetic mechanism may also be involved.

Disrupted immune system metabolism might be a contributing factor to aRFA occurrence. We identified genes involved in the immune system cluster (Figure 5) that are involved in glucocorticoid regulation, enzyme and ion cellular transport, and inflammatory response [www.genecards.org, accessed on 19 February 2022]. While the inflammation seen in the skin biopsies of two individuals is most likely caused by an impaired epidermal barrier and does not have a primary genetic cause, an altered glucocorticoid regulation, enzyme function, or ion cellular transport might be associated with the HC.

When we identified genes (out of the 144 GWAS candidates) connected to the preliminarily chosen metabolic pathways that could have a connection with aRFA, we discovered four genes controlling the circadian rhythm (*RORA*, *PIF1*, *TCF12*, *CSNK2A1*) and four keratin-associated genes (*FUT8*, *ZFP36L1*, *RNF111*, *SNX22*) (Appendix A). The genes controlling the circadian rhythm might be associated with the seasonality of the disease. Besides circadian rhythm metabolism, melatonin metabolism hs also been discussed as one of the causative factors of RFA [27]. In the protein–protein analysis of interacting GWAS genes and DEGs shown in Figure 5, *Kyneurine 3-Monoxygenase* (*KMO*) and its paralog *Coenzyme Q6* (*COQ6*) were identified. Interestingly, *KMO* is part of the nicotinamide adenine dinucleotide (NAD) biosynthesis II (from tryptophan) pathway and the tryptophan utilization superpathway, and is thus directly connected to the melatonin degradation pathway [www.pathcards.genecards.com, accessed on 19 February 2022]. Although it has not been shown for dogs with aRFA, more than 50% of dogs with RFA respond well to melatonin treatment [27]. 

## 5. Conclusions

In this comprehensive study, we investigated the genetics and the differential gene expressions associated with aRFA in the Cesky Fousek using a unique combination of techniques. We performed a genome-wide association study on 216 individuals, RNA-seq experiments from skin biopsies of 11 dogs, and examined the histopathological phenotype of dogs with aRFA. This is the first complex genomic study of canine alopecia in dogs using such an extensive sample size. Histologically, we found that aRFA is similar to RFA and compatible with an impaired HC. The mRNA of genes associated with the initiation and promotion of the HC, as well as of genes encoding for follicular stem cell markers, were mostly downregulated. These findings explain the lack of anagen follicles in the skin of affected individuals. In total, we identified 144 candidate genes from the GWAS analysis (including both the significant and suggestive associations) and 236 strongly deregulated genes from the RNA-seq analysis. Using only the significant GWAS candidate genes (n = 12), we did not discover a direct functional connection to the strongly deregulated genes. However, using the suggestive GWAS candidate genes we discovered four major metabolic pathways associated with aRFA—collagen formation, muscle structure/contraction, lipid metabolism, and the immune system. The findings from our study suggest that aRFA has a complex genetic inheritance that warrants further study.

Given the limitations of the GWAS analyses, further genetic studies involving independent and larger cohorts, including multiple breeds, are needed in order to validate our findings and determine the specific variants that contribute to aRFA risk.

## Figures and Tables

**Figure 1 genes-13-00650-f001:**
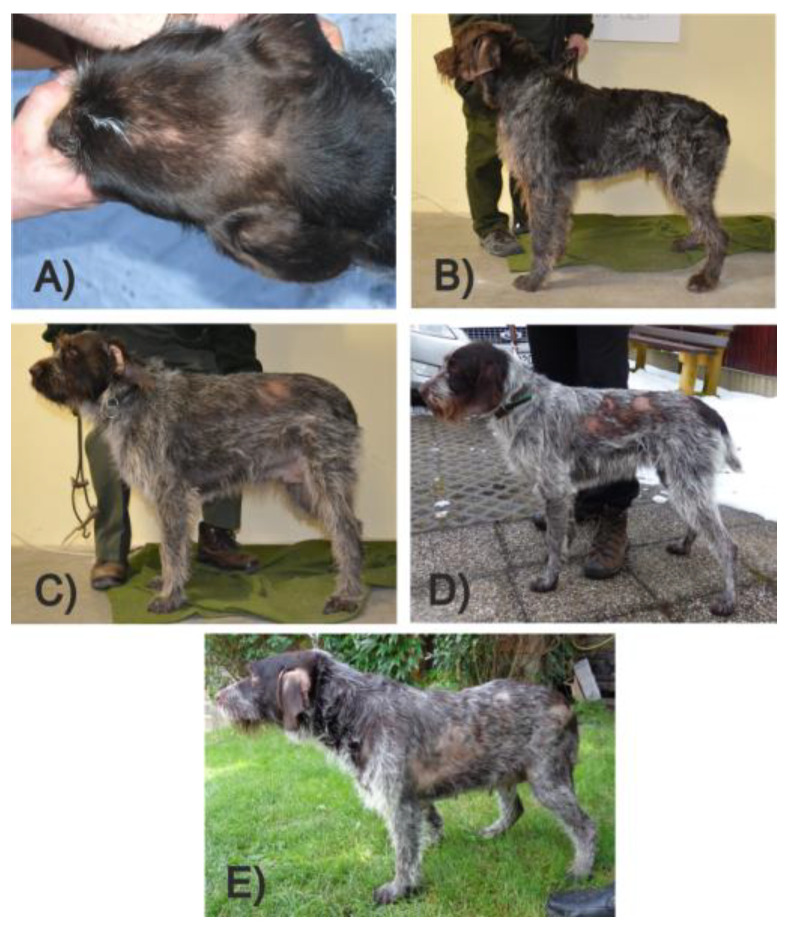
Cesky Fousek individuals affected by aRFA. (**A**) unusual manifestation on the head; (**B**) level 1 aRFA—loss of hair on ears only; (**C**) level 2 aRFA—loss of hair on the body sides up to approximately 10 × 10 cm; (**D**) level 3 aRFA—loss of hair on the body sides up to approximately 10 × 25 cm; (**E**) level 4 aRFA—loss of hair of the body sides up to approximately 10 × 40 cm. Pictures A and E were taken prior to the hair-loss peak in these individuals; alopecia worsened in the weeks after the pictures were taken.

**Figure 2 genes-13-00650-f002:**
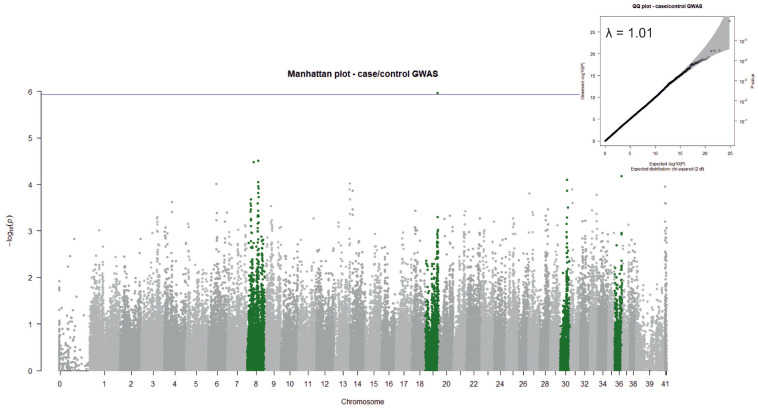
Manhattan and QQ plot for case/control GWAS. The chromosomes of the nine most significant SNPs are shown in green. The significance threshold (shown as a purple line) was set based on Bonferroni correction (cut-off = 1.16 × 10^−6^). The lambda value is shown in the QQ plot.

**Figure 3 genes-13-00650-f003:**
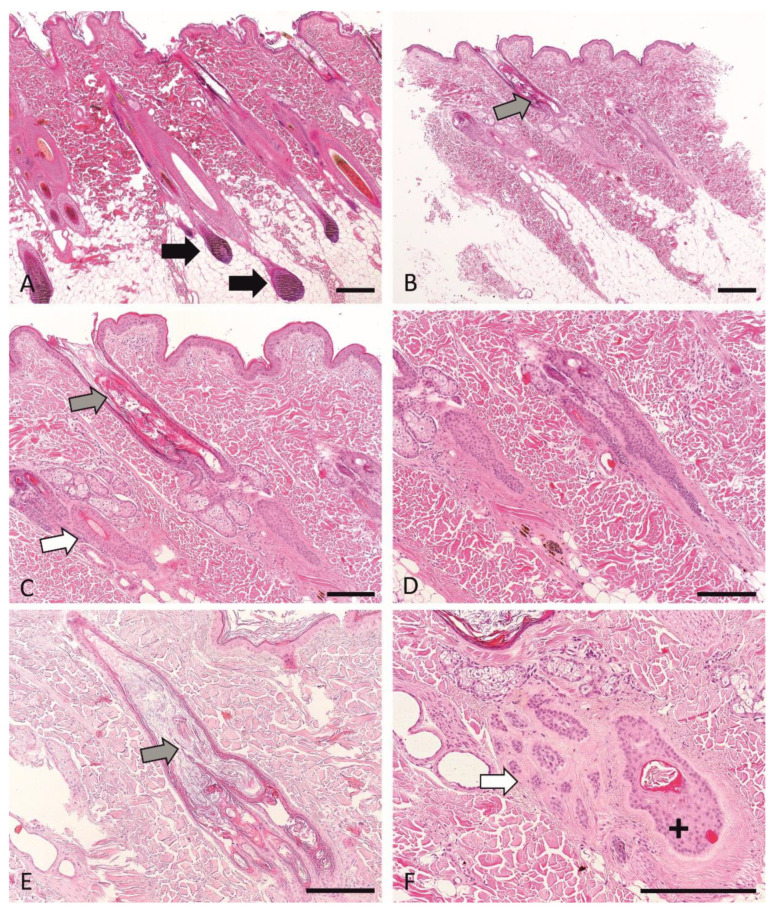
Histological representation of biopsy samples from control skin from unaffected dogs (**A**) and affected dogs (**B**–**F**). Note numerous anagen hair follicles in A identified by the presence of numerous hair bulbs (black arrows). In aRFA, infundibuli (gray arrows) are moderately to severely dilated (**B**,**C**,**E**) and are filled with abundant keratin, which extends into the openings of the secondary follicles, resulting in a “witch’s feet”-like appearance (**E**). The follicular parts proximal to the infundibuli are shortened and limited to the dermis (**B**–**E**). A few telogen follicles (**C**, white arrow) or kenogen follicles (**F**, white arrow) can be identified. Follicular atrophy may be seen (**F**, black cross). A mild distortion of the HFs is observed (**B**–**F**). All samples are stained with hematoxylin and eosin (H&E) and the scale bars represent 200 microns.

**Figure 4 genes-13-00650-f004:**
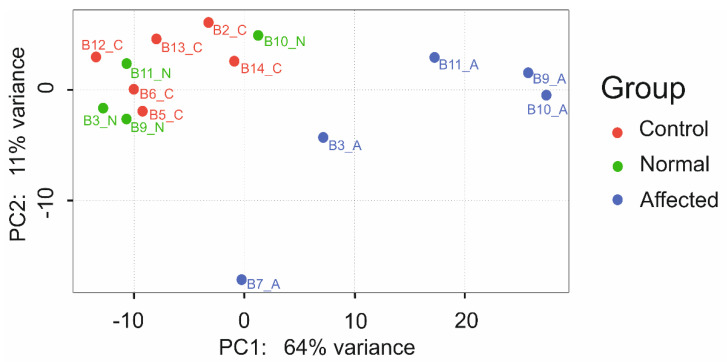
Principal component analysis (PCA) of samples demonstrating clustering based on expression profiles plotted against the two most variable components (PC1 and PC2). Samples from control animals (red) and normal skin from affected animals (green) cluster together, whereas samples from alopecic skin from affected animals (blue) are clearly separated from the clusters representing normal skin but show a higher inter-group variability.

**Figure 5 genes-13-00650-f005:**
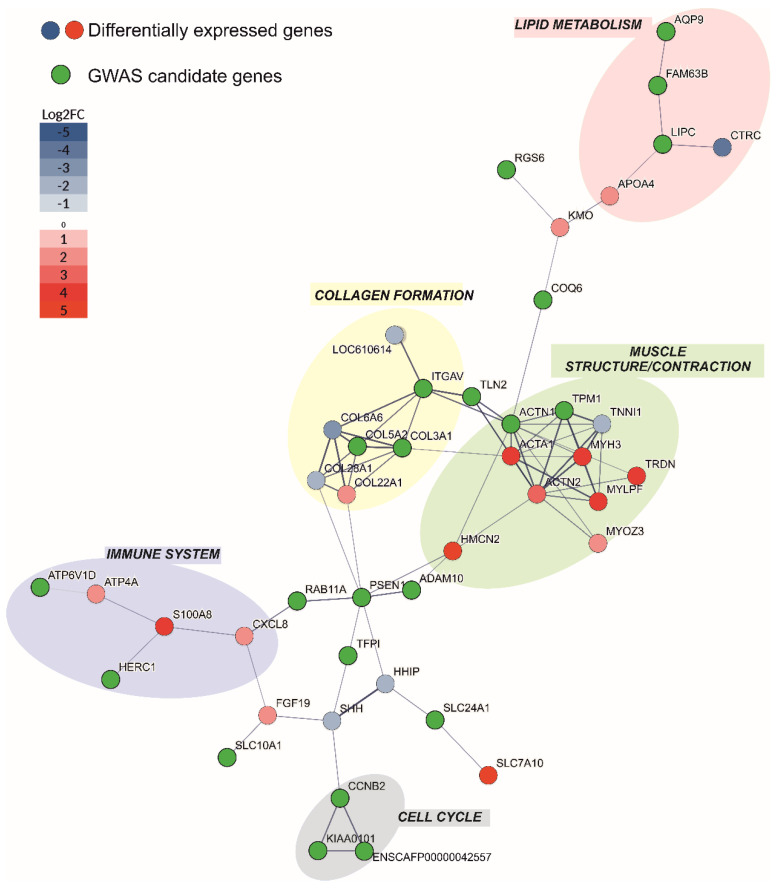
Interactions of GWAS candidate genes (green) and STRING-associated strongly differentially expressed genes colored by their level of expression. We used only medium confidence associations and higher (increasing thickness of lines connecting genes indicates greater confidence). Colorful bubbles represent the metabolic pathways common for each cluster of genes.

**Table 1 genes-13-00650-t001:** Case/control GWAS results showing chromosome (Chr), SNP name, position (bp), allele frequency, and raw *p*-value for the top twenty SNPs. One significant SNP was identified on chromosome 19. The interrupted line represents the Bonferroni cut-off.

Chr	SNP Name	Position (bp)	Allele Freq	*p*-Value
19	BICF2G630255452	47,856,573	0.333	1.08 × 10^−6^
8	BICF2P465820	43,487,284	0.262	3.10 × 10^−5^
8	TIGRP2P114211_rs8542415	434,942,31	0.262	3.10 × 10^−5^
8	BICF2S23110497	25,810,719	0.205	3.30 × 10^−5^
36	BICF2P1194573	28,584,717	0.271	6.72 × 10^−5^
30	BICF2G630401492	26,273,661	0.326	8.07 × 10^−5^
8	BICF2P361090	43,341,287	0.233	8.90 × 10^−5^
8	BICF2P543725	43,371,261	0.233	8.90 × 10^−5^
8	BICF2S23137831	43,418,611	0.233	8.89 × 10^−5^
13	BICF2P281837	63,012,417	0.057	9.61 × 10^−5^
6	BICF2P742566	35,078,147	0.309	9.86 × 10^−5^
8	BICF2P177234	43,520,222	0.235	1.10 × 10^−4^
41	BICF2S23546044	18,45,101	0.310	1.11 × 10^−4^
8	TIGRP2P114933_rs9187625	46,799,348	0.493	1.24 × 10^−4^
31	BICF2P1368177	7,605,782	0.104	1.27 × 10^−4^
31	BICF2S2443709	7,615,165	0.104	1.27 × 10^−4^
13	BICF2G630745860	61,855,230	0.149	1.28 × 10^−4^
14	BICF2G630521203	10,825,554	0.061	1.36 × 10^−4^
30	TIGRP2P370921_rs8763952	26,977,673	0.233	1.36 × 10^−4^
8	BICF2P1102123	43,411,814	0.255	1.54 × 10^−4^

**Table 2 genes-13-00650-t002:** Genotypes of the two top SNPs from the case/control GWAS for chromosomes 19 and 8. The highest proportions in each phenotype group are underlined.

Chr (SNP)	Genotype	No. Controls	% Controls	No. Cases	% Cases
chr19 (BICF2G630255452)	AA	69	59.0	26	26.8
	GA	41	35.0	53	54.6
	GG	7	6.0	17	18.6
chr8 (BICF2P465820)	AA	16	13.7	1	1.0
	AG	50	42.7	27	27.8
	GG	51	43.6	68	70.1

**Table 3 genes-13-00650-t003:** Quantitative GWAS results showing associations with six phenotypic categories. Chromosome (Chr), SNP name, position (bp), allele frequency, and raw *p*-value for the top twenty SNPs. No variant reached the significance cut-off, and thus these variants are considered suggestive only.

Chr	SNP Name	Position (bp)	Allele Freq	*p*-Value
8	BICF2P361090	43,341,287	0.235	4.81 × 10^−6^
8	BICF2P543725	43,371,261	0.235	4.81 × 10^−6^
8	BICF2S23137831	43,418,611	0.235	4.81 × 10^−6^
8	BICF2P465820	43,487,284	0.263	5.56 × 10^−6^
8	TIGRP2P114211_rs8542415	43,494,231	0.263	5.56 × 10^−6^
8	BICF2S23110497	25,810,719	0.207	5.81 × 10^−6^
8	BICF2P177234	43,520,222	0.236	7.36 × 10^−6^
19	BICF2G630255452	47,856,573	0.335	9.05 × 10^−6^
8	TIGRP2P114933_rs9187625	46,799,348	0.495	1.36 × 10^−5^
8	BICF2S23235533	15,314,523	0.251	1.53 × 10^−5^
8	BICF2S22921051	15,005,970	0.260	1.56 × 10^−5^
8	BICF2P1102123	43,411,814	0.256	1.85 × 10^−5^
8	BICF2P1109401	43,462,069	0.256	1.85 × 10^−5^
8	BICF2P146090	43,425,554	0.256	1.85 × 10^−5^
8	BICF2P396875	43,463,543	0.256	1.85 × 10^−5^
8	BICF2P755461	43,441,286	0.256	1.85 × 10^−5^
8	BICF2P762487	43,454,904	0.256	1.85 × 10^−5^
8	BICF2S22932019	46,809,268	0.493	2.18 × 10^−5^
31	BICF2P1368177	7,605,782	0.102	2.58 × 10^−5^
31	BICF2S2443709	7,615,165	0.102	2.58 × 10^−5^

**Table 4 genes-13-00650-t004:** Genotypes for the top SNP (chr8, BICF2P361090) from the QGWAS for each of the six phenotypic categories, with percentages shown in parentheses. The highest proportions in each phenotype group are underlined.

Chr (SNP)	Genotype	Healthy (%)	Head (%)	L1 (%)	L2 (%)	L3 (%)	L4 (%)
chr8 (BICF2P361090)	AA	52 (47)	1 (33)	4 (67)	14 (50)	43 (88)	15 (79)
	CA	46 (41)	2 (67)	1 (17)	13 (46)	5 (10)	4 (21)
	CC	13 (12)	0	1 (17)	0	1 (2)	0
	missing	0	0	0	1 (4)	0	0
**total**		111	3	6	28	49	19

**Table 5 genes-13-00650-t005:** Additional analysis (age of onset before 2 years of age) GWAS results showing chromosome (Chr), SNP name, position (bp), allele frequency, and raw *p*-value for the top twenty SNPs. One significant SNP was identified on chromosome 21. The interrupted line divides significant association from the rest. The significance threshold based on the Bonferroni correction was set to 5.8 × 10^−7^.

Chr	SNP Name	Position (bp)	Allele Freq	*p*-Value
21	BICF2G630640798	47,085,771	0.221	5.01 × 10^−7^
23	BICF2S23432401	11,113,618	0.377	5.75 × 10^−6^
37	TIGRP2P420015_rs8709645	20,114,103	0.262	6.48 × 10^−6^
8	chr8_59707832	59,707,832	0.221	8.66 × 10^−6^
37	BICF2G630131116	25,669,986	0.148	1.07 × 10^−5^
15	BICF2G630419811	59,659,531	0.434	1.22 × 10^−5^
23	BICF2P438054	11,110,146	0.352	1.30 × 10^−5^
21	BICF2G630641744	46,513,869	0.254	1.60 × 10^−5^
27	BICF2G630139626	42,94,734	0.205	1.92 × 10^−5^
17	chr17_40427743	40,427,743	0.426	2.87 × 10^−5^
21	BICF2S23427379	46,584,445	0.270	3.07 × 10^−5^
20	BICF2P1328442	55,962,058	0.320	3.49 × 10^−5^
27	BICF2P675588	34,378,821	0.295	3.70 × 10^−5^
23	BICF2G630386401	13,368,971	0.459	4.20 × 10^−5^
18	BICF2G630699395	34,387,737	0.484	4.30 × 10^−5^
27	BICF2G630139599	4,253,386	0.180	4.40 × 10^−5^
27	BICF2G630139609	4,266,185	0.180	4.40 × 10^−5^
27	BICF2G630139630	4,299,688	0.180	4.40 × 10^−5^
27	BICF2G630139642	4,318,805	0.180	4.40 × 10^−5^
27	BICF2S23028384	4,247,215	0.180	4.40 × 10^−5^

**Table 6 genes-13-00650-t006:** Haplotypes for chromosomes revealed by the case/control and quantitative GWAS and subsequent haplotype analysis. For each chromosome we show the most significant haplotype and a haplotype containing the most significant or suggestive SNP for each chromosome (marked by *).

Chr	bp	Haplotype	% Cases	% Controls	*p*-Value
8	43,341,287–43,356,221	AAG	75.0	49.4	7.31 × 10^−8^
8 *	43,463,820–43,494,231	GGG	85.0	62.4	2.25 × 10^−7^
19	19,807,697–20,172,164	ATGGTCAGGG	84.4	53.9	2.09 × 10^−11^
19 *	47,856,573	A	54.7	76.5	2.03 × 10^−6^
19 *	47,856,573	G	45.3	23.5	2.03 × 10^−6^
30	26,126,946–26,143,675	GCGA	15.8	35.5	5.04 × 10^−6^
30 *	26,245,545–26,328,881	ATACAGGA	21.5	41.3	1.45 × 10^−5^
36 *	28,573,704–28,584,717	CC	36.7	18.8	3.53 × 10^−5^

**Table 7 genes-13-00650-t007:** Differentially expressed genes associated with a role in hair follicle (HF) morphogenesis and the hair cycle (HC) identified by transcriptome analysis comparing unaffected skin of control dogs and dogs affected with aRFA (n = 11) with alopecic skin of dogs with aRFA (n = 5).

Gene Symbol	Full Gene Name	Described Function	Signaling Pathway	BaseMean	Log2FC
*CTNNB1*	catenin beta	promotes HF growth	WNT	20,928.210	−0.498
*CUX1*	Cutl1, cut like homeobox 1	inhibitor of HF differentiation	NOTCH	1600.827	−0.895
*DLX1*	distal-less homeobox 1	HF cycling and differentiation	WNT	48.951	−2.120
*DLX2*	distal-less homeobox 2	HF cycling and differentiation	TGF-b	32.891	−1.852
*DLX3*	distal-less homeobox 3	HF cycling and differentiation	WNT	1315.187	−1.154
*DLX5*	distal-less homeobox 5	HF cycling and differentiation	BMP	85.608	1.026
*FGF5*	fibroblast growth factor 5	catagen induction	FGF	124.674	−2.896
*FOXE1*	forkhead box E1	governs HF stem cell (SC) niche	SHH	475.559	−1.575
*FOXN1*	forkhead box N1	HF development, HS differentiation	WNT, BMP, SHH	1199.807	−1.476
*FZD2*	frizzled class receptor 2	receptor WNT pathway	WNT	155.689	−0.948
*FZD3*	frizzled class receptor 3	receptor WNT pathway	WNT	412.223	−0.978
*GLI2*	GLI family zinc finger 2	HF SC related transcription factor	SHH	541.121	−0.927
*HHIP*	hedgehog interacting protein	HF organogenesis	SHH	185.270	−2.185
*HOXC13*	homeobox C13	HS differentiation	WNT	1064.553	−1.734
*JAG1*	Jagged 1	HF maintenance	Notch	5310.336	−0.668
*LEF1*	lymphoid enhancer binding factor 1	HS differentiation	WNT	957.142	−1.636
*LGR4*	leucine rich repeat containing G protein-coupled receptor 4	delays HC; inhibits activation of follicular SCs	WNT	4786.889	−0.464
*LGR5*	leucine rich repeat containing G protein-coupled receptor 5	follicular SC marker; anagen initiation	WNT	1028.425	−1.430
*LGR6*	leucine rich repeat containing G protein-coupled receptor 6	SC associated marker	WNT	336.586	1.145
*LHX2*	LIM homeobox 2	HF differentiation, SC associated marker	WNT	645.982	−1.377
*MSX2*	Msh homeobox 2	HS differentiation	BMP	295.450	−1.454
*NCAM1*	neural cell adhesion molecule 1	expressed in dermal papilla	FGF	267.400	−1.642
*NFATC2IP*	nuclear factor of activated T cells 2 interacting protein	aging of HF stem cells		1056.715	0.332
*SHH*	Sonic hedgehog	HF development and cycling	SHH	51.377	−2.002
*SMO*	Smoothened	HF development and cycling	SHH	781.216	−0.858

**Table 8 genes-13-00650-t008:** Differentially expressed genes associated with either vitamin D or steroid hormone metabolism comparing unaffected skin of control dogs and dogs affected with aRFA (n = 11) and alopecic skin of dogs with aRFA (n = 5).

Gene Symbol	Full Gene Name	Function	BaseMean	Log2FC
*CYP27B1*	cytochrome P450 family 27 subfamily B member 1	activates vitamin D3	209.288	−1.650
*CYP2R1*	cytochrome P450 family 2 subfamily R1	major vitamin D25-hydroxylase	228.196	0.980
*CYP39A1*	Cytochrome P450 Family 39 Subfamily A Member 1	7-alpha hydroxylation of 24-hydroxycholesterol	753.872	−0.818
*CYP51A1*	cytochrome P450 family 51 subfamily A member 1	cholesterol biosynthesis	1933.523	0.712
*DHCR7*	7-Dehydrocholesterol reductase	converts 7-dehydrocholesterol (substrate for vitamin D formation cholesterol)	2100.759	0.719
*ESR2*	estrogen receptor 2	nuclear receptor, expressed in the HF in outer root sheath, dermal papilla, matrix cells and in the bulge	180.354	−1.212
*HSD17B2*	17β-Hydroxysteroid dehydrogenase 2	inactivation of estrogens and androgens: converts estradiol to estrone, testosterone to androstenedione, and androstenediol to DHEA; activates the weak progestogen 20α-hydroxyprogesterone into the potent progestogen progesterone	2720.505	1.061
*HSD17B6*	17β-Hydroxysteroid dehydrogenase 6	androgen catabolism: convert 3 alpha-adiol to dihydrotestosterone and androsterone to epi-androsterone.	310.914	0.762
*HSD17B7*	17β-Hydroxysteroid dehydrogenase 7	biosynthesis of estrogen and cholesterol	1341.347	0.687
*HSD3B2*	hydroxy-delta-5-steroid dehydrogenase, 3 beta- and steroid delta-isomerase 2	biosynthesis of all classes of hormonal steroids	154.717	2.406
*RXRG*	retinoid X receptor gamma	increases transcriptional function of VDR	147.075	−1.132
*VDR*	vitamin D receptor	nuclear transcription factor, absence leads to defects in HF regeneration and alopecia	3347.023	−1.211

## Data Availability

The data will be available on the Dryad platform (https://datadryad.org/stash/ accessed on 19 February 2022) if this paper is accepted.

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
