# Peer review of "Genomic and Transcriptomic Characterization of Atypical Recurrent Flank Alopecia in the Cesky Fousek"

_genes, 2022, doi:10.3390/genes13040650_

Round 1

Reviewer 1 Report

This was a well designed study, adeptly analyzed, with findings relevant both to veterinarians and more broadly of interest in understanding the genetic foundations of alopecia.

My only feedback on this excellent paper is in its discussion/conclusions, and a related piece of background information that would belong in the introduction. As the authors concluded, aRFA is clearly polygenic. However, it is very likely that there is a genetic predisposition to aRFA that is fixed in the Cesky Fousek breed, and that the markers identified in this GWAS are modifiers of that predisposition - perhaps not even with different allele frequencies in this breed than in others. This possibility is not discussed in the conclusions, and given the breed predisposition of the disease and the lack of strong association findings, I think it's important to discuss.

Compare to

Tang, Ruqi, et al. "Candidate genes and functional noncoding variants identified in a canine model of obsessive-compulsive disorder." Genome biology 15.3 (2014): 1-15.

which assessed OCD in Doberman Pinschers (DP) and other breeds: "The DP breed, like all dog breeds, was created through population bottlenecks and artificial selection for morphological and behavioral traits, potentially driving some OCD risk alleles to very high frequency and thus undetectable by GWAS. Consistent with this hypothesis, we find functional connections between associated genes and genes in the 13 largest autosomal regions of fixation in the DP breed..."

It is important to note that future studies may need to take a similar approach of cross-breed comparisons in order to identify a potentially fixed risk factor in the Cesky Fousek. Additionally, the potential for a fixed risk factor will be of interest to breeders of the Cesky Fousek, who may use it to consider the utility of an outcross program.

Related:

  • Some discussion of the actual incidence of this disease in the Cesky Fousek should be in the introduction. If reliable numbers don't exist, some context for the statement that the disease has a "high frequency" in the breed is helpful.
  • Given the likelihood that the most strongly associated markers you have found may be used in a future genetic test, it is the responsible thing to do to specifically state that these markers are not predictive of the disease. Statistics to illustrate this statement would be particularly useful.

Reviewer 2 Report

The current manuscript presents genome wide association and RNAseq analysis data that attempts to identify a genetic bases and potential mechanism for atypical recurrent flank alopecia in the Cesky Fousek breed.  The authors' approach is logical and the manuscript is well written with appropriate references.

The presented GWAS data, however, does not appear to have statistically significant value.   The authors use a very high FDR of 20% (p < 0.2), which seems arbitrary and can lead to a significant number of false positive results.  Furthermore, only 1 SNP (on Chr19) was above the set significance threshold (genome wide significance), while all the others were well below it.  Yet the authors present a list of 143 “candidate genes”, of which only 11 are located on Chr19 and the rest found near SNPs that were not significantly associated with aRFA.  Presenting the data as it is and acknowledging the limitations and lack of significance would still be appropriate, except the authors then use the data to make further associations and conclusions regarding potential protein interactions and metabolic pathways associated with aRFA.   These conclusions are not supported by the GWAS data.  Statistically, the authors could run the exact same analysis using a different population set and would likely come up with a completely different set of candidate genes, with different potential protein interactions, and a whole different set of associated pathways.

The RNAseq data is interesting and does hold value, although it is concerning that the affected aRFA samples do not appear to cluster well on the PCA plot.  The normal samples cluster along PC2, but the affected dogs are spread across both PC1 and PC2.   This suggests potentially significant expression differences between the affected samples and possibly different mechanisms underlying the pathology between patients.  Regardless, the RNAseq data does present significantly differentiated genes unique to the diseased tissue and analysis of those genes as they relate to HC and HF morphogenesis, as well as correlation to histopathological findings, is novel and of scientific value.  That being said, the comparisons made between the current RNAseq data and the author’s previously published data from reference #56 needs to be re-examined.  Different types of comparisons were used for the differential gene expression analyses performed in each study (normal v affected here, and telogen v anagen in #56) and as such, direct comparison between the DEGs of both studies is confounded by those differences. 

I recommend a major revision of the manuscript that highlights the limitations of the GWAS data and associated downstream analysis.  In addition, correlations between the non significant GWAS data and the RNASeq data should be limited, which it already is because no significant direct correlations were observed.  Furthermore, the protein – protein interaction analysis data is completely hypothetical, essentially an association (predicted protein interaction), of an association (gene), of an association (SNP), that lacks statistical relevance.  Conclusions regarding a potential pathways contributing to aRFA pathology should focus on the RNAseq data, for which significance was observed.

Specific Comments:

Materials and Methods:

Blood Sample collection:

  1. What was the duration of the study period?

  1. What was the age of the dogs at collection? How old are the dogs now?

  1. What was the criteria for determining the control dogs? If young dogs were used, how could the authors be sure those control dogs would not develop aRFA at a later age, thus confounding the GWAS analysis? As the authors point out in the introduction, aRFA presents later in life. 

Biopsy sample collection:

  1. The authors describe 7 control dogs and 7 affected dogs for biopsy collection for the RNAseq analysis. The “comprehensive list of animals used for this study” found in Table S2 only lists 11 dogs. The histopathological analysis section later explains that 1 control and 2 alopecia dogs were excluded based on histopathology.  This explanation should be mentioned in the biopsy sample collections section, or table S2 should be edited to actually be “comprehensive” and include all dogs and indicate which were included and excluded.

Genotyping

  1. What was the reasoning for using an FDR of 20%?

  1. What was the p-value threshold for genome wide significance?

  1. Are the presented p-values raw p-values, or corrected (Bonferroni)?

QGWAS

  1. How were the phenotypic categories determined? Were they based on clinical diagnostic criteria?

  1. The use of only control dogs aged 10+ years “in which the chance of developing aRFA is very low” is appropriate here. However, this suggests that the other 80 control dogs used in the case/control GWAS were younger and did have a chance of developing aRFA later in life. Did the authors follow up on all the control dogs until they all reached 10 years of age in order to confirm no controls developed aRFA?  If not, this is a significant design flaw.

Differential expression analysis

  1. Line 251-252- Please provide a reference for Reactome.

  1. Line 253 – “A list of upregulated and downregulated genes was uploaded separately into the database and was analyzed and matched with known biological processes and pathways.” Do the authors mean separate lists of upregulated and downregulated genes were uploaded, or a single list with both downregulated and upregulated genes? Please clarify.  If separate lists were used, were any identified pathways common to both lists (results question)? 

Protein-Protein interactions:

  1. Line 259 – Can the authors clarify what they mean by “We were able to retrieve predicted protein-protein interactions for 132 out of 143 GWAS candidates and for 144 out of 236 strongly DEG (exceeding the Log2FC value of +/-2).” Were they only able to get protein information from the STRING database for 132 GWAS and 142 DEGs, respectively (meaning only protein data from 132 GWAS and 142 DEGs were available for the analysis)? Or do they mean they identified protein-protein interactions between the 132 of the GWAS genes themselves, and similarly between 144 of the strongly DEGs?

Results

  1. Lines 416-417 – “A PCA plot was constructed based on gene expression profiles (controls n=6; affected n=5) and demonstrates distinct clustering of samples from control dogs and biopsies of unaffected skin from affected dogs compared to alopecic skin of dogs with aRFA (Figure 4).” The authors left out the “normal” dogs sample number – (Controls n=6, NORMAL n=4, affected n=5). Then later when describing the combination of the controls and normal dogs, the authors have n=11, but there are only 10 dogs in those combined categories in the PCA plot: 6 control, and 4 normal.

  1. Figure 4. It does not look like the affected skin samples clustered together at all – Blue dots.

Comparison of deregulated genes

  1. The author’s previously published data from reference #56 is used for making comparisons with the current data and are included in the results section. The methods used for obtaining the RNAseq data from microdissected telogen HF and anagen HF from ref 56 should be briefly described in the methods of the current paper. This is very important considering different types of samples (including different breeds) were used for the previous paper and different types of comparative analysis were made and the methods used to filter, align, and normalize data may be very different.

  1. Paragraph 457 – The authors are attempting to make comparisons between the DEGs in the current study to the baseline gene expression in telogen HF anagen HF reported in ref #56.

I do not understand the relevance of this comparison.  The DEGs reported here are upregulated or downregulated relative to normal, non-affected skin, and the genes expressed in telogen HF anagen HF in ref #56 were simply expressed, not relative to any type of control.    By only looking at the DEGs, the authors ignore genes equally expressed in both normal HF and affected HF.   In addition, and as noted in the results, only a very small percentage of the DEGs were expressed solely in anagen (2.2%) or telogen (0.7%).  As such, the significance of the differential gene expression of these few genes to the overall aRFA phenotype is minimal, relative to the other 1,394 identified DEGs.  A better comparison would be to perform a cluster analysis of the aRFA, anagen, and telogen baseline gene expression data and see which groups cluster together. 

  1. Paragraph 470 - The authors are attempting to make comparisons between their differential gene expression in which aRFA samples were compared to normal samples and ref 56 differential gene expression in which telogen samples were compared to anagen samples. These are 2 different DEG data sets, in which the differential expressions are relative to different samples, and therefore, these 2 sets of DEG data cannot be compared in this way. If the telogen samples and anagen samples in ref #56 were each compared to “normal” or total HF gene expression, that would be different as the “control” samples would be similar between studies.

  1. In ref #56, 2 main analysis were performed: 1. Looking at baseline gene expression in telogen HF, anagen HF, and interfollicular epidermis (IFE). In that analysis, ref 56 simply looked at genes that were expressed and not differentially expressed – there was no DEG comparison made between groups.  In the second analysis of Ref 56, differential expression between the telogen and anagen HF were made, and in this case, the expression values are relative to each other, meaning that genes upregulated in telogen were upregulated relative to anagen, and therefore the same genes would be downregulated in anagen, relative to telogen. 

In the current data results section, however, the authors make the following statement:

Line 472 - “The majority of genes that were upregulated in the biopsies of alopecic skin from dogs with aRFA were downregulated in microdissected telogen HFs (n = 24) and those that were downregulated in dogs with aRFA were upregulated in microdissected anagen HFs (n = 124) (Table S9).”

That statement is completely contradictory.  The identified downregulated genes in telogen from ref #56 are, by nature of the analysis, upregulated in anagen, because the DEGs in telogen are relative to anagen.  Therefore, if the genes upregulated in the current study from biopsies of aRFA dogs were downregulated in microdissected telogen (relative to anagen), those same genes would, by nature of the analysis, be upregulated in microdissected anagen (relative to telogen).  Therefore, if the first part of the statement (line 472) is true, then the second part of that statement in which “those (genes) that were downregulated in dogs with aRFA were upregulated in microdissected anagen HFs” simply cannot be true.

  1. In addition, there is a discrepancy in this same data between the results section and the discussion section regarding downregulated and upregulated genes in aRFA compared to DEG in anagen and telogen from ref 56:

Results Line 472:  “The majority of genes that were upregulated in the biopsies of alopecic skin from dogs with aRFA were downregulated in microdissected telogen HFs (n = 24) and those that were downregulated in dogs with aRFA were upregulated in microdissected anagen HFs (n = 124) (Table S9).”

Discussion Line 569:  “Comparing the deregulated genes of dogs with aRFA with those of anagen and telogen microdissected HFs revealed that the downregulated genes in alopecic skin from dogs with aRFA are upregulated in microdissected anagen HFs and the upregulated genes in alopecic skin biopsies from dogs with aRFA are downregulated in microdissected anagen HFs.”

Which is correct?  This is very confusing.  The description in the results (which appears contradictory) would suggest that the aRFA DEG signature does not match either telogen or anagen, while the description in the discussion suggests the aRFA DEG signature is not like anagen, with minimal similarities to telogen.  What about catagen?  The discussion seems to focus on the RNA signature being either anagen or telogen, but nothing about catagen.  Is that because there is no catagen RNAseq data for comparison?

  1. Table S6 – The authors should add reference 56 in the figure legend as they did in S7, S8, S9 table captions.

  1. Line 499 - The authors make the following statement: “Due to the lack of overlap between GWAS candidates and DEGs, we speculated that they may be linked at the level of functional processes. This would mean that genetic changes identified in GWAS analysis are resulting in gene expression shifts of the interacting protein partners.”

Can the authors please explain what this statement means?  What explanation do they offer that might explain how “the genetic changes identified in GWAS” might result “in gene expression shifts of the interacting protein partners”?   Are the authors suggesting that the genetic changes identified in the GWAS resulted in changes in the protein of one or more of the candidate genes, and that altered protein then differentially influenced transcription or RNA stability of the DEGs?  If so, are any of the GWAS candidate genes described in Figure 5 known to regulate gene expression, are any transcription factors or similar types of proteins? 

  1. Figure 5. It is very difficult to read the gene names and also very difficult to discern between the thickness of the lines which corelate with greater association confidence.

Discussion

  1. Line 534 – Do the authors mean chr19 and chr8 (NOT chr21) – from Table 2? Or are they referring to the significant SNP identified on chr21 from the additional analysis comparing aRFA onset before 2 years of age and level 4 aRFA from Table 5? If so what about the variant from Chr8 listed in table 2?

  1. The Paragraphs beginning with line 533 and line 545 seem to discuss the same data, but in very different ways. The first paragraph discusses the 2 significant associations identified from the GWAS data (although I think it should be 19 and 8, not 21). The authors then talk generally about the strengths and limitations of the GWAS and the 2 identified associations.  The second paragraph introduces 143 GWAS candidate genes, identified on several chromosomes, but then goes on to focus on the details of the Chr21 and Chr8 SNPs.  It’s not clear if the authors are talking about different data (SNPs) in the 2 paragraphs, or the same data/SNPs.  If they are the same, the paragraphs should be merged to be more congruent.

  1. In addition, several different GWAS analysis were performed and described in the results and several significant SNPs are mentioned for chr19 and for chr8 (Tables 1-5). When discussing the frequency of the different variants identified on Chr19 and Chr8 in the discussion, it would be helpful to indicate either the SNP ID (in parenthesis) or the bp location of those specific variants in the text for clarification.

  1. Line 541 – how is human genome region hg38 related to canine chr19? This topic needs elaboration.

  1. Line 572 – “These data assign a clear telogen signature to dogs with aRFA although histologically telogen follicles are not the most prominent feature and short strands of well-developed follicular epithelium are predominating.” I agree with the authors later statement (line 593-4) regarding the aRFA transcriptome data supporting a non-anagen signature that aligns with the lack of anagen HFs observed in the histopath. However, I strongly disagree that the RNAseq data assigns a “clear telogen signature to dogs with aRFA”.  Is that statement based on the 11 upregulated genes accounting for only 0.7% of all DEGs in the aRFA samples that were only expressed in telogen HF from ref #56?  Are the authors suggesting that those 11 DEGs are driving the phenotypic signature of aRFA skin?  A strong argument could be made that the RNAseq data show neither an anagen nor a telogen signature.  What about catagen?  Or perhaps the gene deregulation results in a completely dysfunctional HC and thus no clear signature.

  1. Line 621 – “Interestingly, also in skin biopsies of dogs with aRFA seven deregulated genes are encoding proteins, mainly enzymes, involved in sex hormone or cholesterol biosynthesis.” Could this observation be attributed to nearly all the biopsy samples coming from female dogs?

Conclusion section

  1. The authors write “Analysis of both sets of candidate genes discovered four major metabolic pathways significantly associated with aRFA - Collagen Formation, Muscle Structure/Contraction, Lipid Metabolism, and Immune system. These findings confirm that aRFA is a polygenic disorder.”

Exactly which data show that the four metabolic pathways were “significantly associated” with aRFA?  The GWAS data only shows significance of the specific SNPs, not the genes themselves.   Only one SNP was above the significance threshold set at 20%, and that SNP was on Chromosome 19, in which only 11 of the 143 GWAS candidate genes were located near, and none of those 11 genes were among the 40 identified to interact with the 236 strongly DEGs shown in Figure 5.  The methods the author’s use to make a connection between aRFA and the four “major metabolic pathways” is extremely weak, essentially an association (predicted protein interaction), of an association (gene), of an association (SNP), from which the initial SNP associations were not genome wide significant using the authors’ own arbitrarily high 20% FDR.  Based on the data presented, the significance of those four pathways to aRFA is no more than the significance of 4 randomly selected pathways of the 27 listed in Reactome.

Round 2

Reviewer 2 Report

The authors have made appropriate changes to the manuscript to highlight the limitations of the GWAS data presented.  In several areas of the text they have changed the wording to describe the non-significant SNP associated genes as “suggestive associations”, although they go on to state that they believe those non significant associations are “relevant as well”.  However, the authors did not alter any of the conclusive statements at the end of the manuscript which claim to have found several “significant” associations when the data were combined, despite their own acknowledgement that all of those associations were “suggestive”.

The authors added the sentence “Although, there are several concerns about GWAS in general, it is still considered the best method for detecting causative variants of hereditary diseases [61].”  This statement in inaccurate.  GWAS itself does not identify causative variants, it identifies ASSOCIATIONS with SNPs.   Those associations can identify potential candidate genes that can be further studied and potentially lead to identification of a causative genetic variant.  Although the authors did perform appropriate RNAseq analysis to help further home in on a potential causative gene through comparative analysis, they simply did not find any. 

Despite this, the authors arbitrarily decided their GWAS candidates were still relevant and continued to perform analysis until they found a link between the GWAS and RNAseq and further make conclusions based on that analysis:

“Analysis of both sets of candidate genes discovered four major metabolic pathways significantly associated with aRFA - Collagen Formation, Muscle Structure/Contraction, Lipid Metabolism, and Immune system. These findings confirm that aRFA is a polygenic disorder.”

However, as noted in my first review, NONE of the significantly associated (Chr19) GWAS candidate genes were found to interact with the significantly differentiated genes from RNAseq.  Only the “suggested associated” genes were and, therefore, the author’s conclusion should also be considered “suggestive”.

Furthermore, the presented data do not “confirm that aRFA is a polygenic disorder”.  The authors mistakenly conclude that because their analysis did not identify a single significant causative variant, then the disorder must be polygenic.   This is illogical reasoning and just poor science.

The “conclusions” section needs to be edited to convey the limitations of the data.
